# FEW-SHOT STYLE-CONDITIONED LLM TEXT GENERATION VIA LATENT INTERPOLATION

## ABSTRACT

We propose a novel, model-agnostic approach for adapting large language models (LLMs) in a few-shot manner to arbitrary styles using text samples from a given author. Rather than predefined features, our method defines style in terms of LLM model weights and uses a variational autoencoder (VAE) to construct a latent space of these weights, allowing for a generic style representation. Our approach leverages interpolation in this latent embedding space of model weights to generate novel finetuned models for low-resource authors. We evaluate this approach compared to reported results, finetuning, and prompting across three datasets. Results indicate that our method outperforms our baselines in low-resource settings.

## 1 INTRODUCTION

Text generation is a core natural language processing (NLP) task that enables applications such as machine translation, summarization and question-answering (Li et al., 2022). Guiding this task such that it outputs text satisfying a certain set of constraints is referred to as controllable text generation (Zhang et al., 2023). One such form of control is style-conditioned text generation i.e., constraining the generated text such that it follows a certain writing style, usually corresponding to a specific author (Mou & Vechtomova, 2020). When generating text using large language models (LLMs), style conditioning is particularly challenging due to the prohibitive training data requirements of LLMs (Zhang et al., 2023). Successfully conditioning the style of text generated via LLMs using just a few samples holds potential for new applications such as real-time style adaptation and empowering users to produce text matching their own writing style.

Existing methods for controlling text generated via LLMs include prompting, finetuning and post-processing (Zhang et al., 2023). Among these, prompting is the least computationally demanding, but prior work has shown that it is incapable of reliably inferring style from a few samples (Patel et al., 2022; Liu et al., 2024). On the other hand, finetuning, while theoretically capable of adapting a model to arbitrary styles, typically requires a sizeable corpus, even when combined with techniques such as low-rank adaptation (LoRA) (Hu et al., 2021). Finally, postprocessing methods which modulate the output probabilities of an LLM can condition the style of generated text in a few-shot manner (Khan et al., 2023). However even this reduced training corpus might be burdensome in applications where a user produces text in real-time. Additionally, many of these methods assume the existence of predefined style features such as, for example, punctuation frequency, ratio of upper-case to lower-case letters and n-gram counts (Lagutina et al., 2019).

In this work, we propose a novel model-agnostic approach for performing few-shot adaptation of an LLM to a target text style. Our approach differs from prior methods by representing style in terms of model weights rather than predefined features and by using a variational autoencoder (VAE) to construct a latent space encoding differences in model weights, which we refer to as *weight deltas*, which we extract using LoRA. As a result, our approach does not assume the availability of predefined style features and leverages the VAE to represent a space of possible finetuned models. We argue that this enables our approach to be more adaptable in terms of style representation. Given a small number of samples from an author, we generate new model weight deltas by performing interpolation in the VAE latent space. We identify two major contributions: (1) a novel model-agnostic method for performing few-shot stylized text generation and (2) a new approach for directly representing text style in terms of model weights.

## 2 RELATED WORK

### 2.1 STYLE TRANSFER AND STYLE-CONDITIONED TEXT GENERATION

Text style transfer (TST) refers to the task of converting a given piece of text from its source style to a target style while preserving its content (Fu et al., 2017; Jin et al., 2022). TST can be categorized into two broad types—attribute-based and authorial. Attribute-based methods aim to transform text along one or more explicitly defined stylistic dimensions (Subramanian et al., 2018; Subramani et al., 2022). Such approaches are limited due to their reliance on labeled data and their inability to model complex styles. Conversely, authorial style transfer (Jhamtani et al., 2017; Syed et al., 2019) aims to transform text to a style that is not straightforward to define explicitly, such as styles attributed to unique authors. Related to but distinct from authorial style transfer, style-conditioned text generation refers to the task of generating text in the style of a given, target author (Tikhonov & Yamshchikov, 2018). This differs from style transfer in that the goal is not to preserve content while transferring style and thus does not require disentangling style from content. Subramanian et al. (2018) demonstrated that this disentanglement, besides being challenging, is unnecessary to model style. The standard approach for this task is to train a language model on a corpus of text in the target style (Tikhonov & Yamshchikov, 2018). However, for transformer-based LLMs, the data requirements for this task have become increasingly prohibitive Zhang et al. (2023). Our work addresses this limitation as we aim to model arbitrary text styles in a few-shot setting while leveraging the generation capabilities of modern LLMs.

### 2.2 FEW-SHOT STYLE-CONDITIONED TEXT GENERATION

LLMs have shown strong capabilities in multiple NLP tasks in both zero-shot and few-shot settings (Zhang et al., 2023). STYLL (Patel et al., 2022) demonstrates that LLMs are able to perform style transfer with arbitrary styles via prompting. However, this approach prompts the LLM to classify the target style using specific attributes which are later used to perform style transfer, similar to work by Reif et al. (2022). This approach performed satisfactorily for attribute-based style transfer but did not demonstrate the ability to model complex authorial styles. Liu et al. (2024) address these issues by proposing ASTRAPOP, an RL-based actor-critic approach to style transfer. However, while this does well on medium-sized corpora, its performance is inconclusive on smaller corpora belonging to a single author. Instead of prompting, Subramani et al. (2022) extract latent vectors from a pretrained LLM which produce desired target sentences when added to the model's hidden states. However, obtaining vectors corresponding to a specific style requires separately extracting vectors for each sentence in that style. Similarly to our own work, Jin et al. (2024) make use of LoRA-based weight increments associated with particular style features, though we do not predefine style features. Finally, Khan et al. (2023) proposed StyleMC, a unified approach to style transfer and stylized text generation using future discriminators (Yang & Klein, 2021). StyleMC relies on pretrained style embeddings in its operation. By instead using model weights directly, we forego the need for any predefined notions of style. We compare our approach to StyleMC in our experiments due to its published results on known datasets. We note that in all of this prior work, even instances focused on few-shot settings, assume access to sixteen or more samples. This data requirement can sometimes exceed hundreds of samples, proving a burden on end users of the approach.

### 2.3 LOW-RANK ADAPTATION (LORA)

Low-Rank Adaptation (LoRA) (Hu et al., 2021) falls under a class of finetuning approaches referred to as parameter efficient fine-tuning (PEFT) methods (Houlsby et al., 2019) which freeze the original weights of a given pretrained model and instead add a smaller set of trainable task-specific weights termed *adapters* to certain layers of the model. In LoRA, this takes the form of passing a given $d$-dimensional input vector $x$ simultaneously to the frozen pretrained layer with weights $W$ and to a learnable projection matrix $A$ which maps $x$ into an $r$-dimensional space where $r << d$. A second learnable matrix $B$ then maps the input back to the $d$-dimensional space and this output is then summed with the output of the frozen layer to produce the output fed to the next layer. We rely on LoRA to make the learned space of weight deltas compact enough to afford interpolation.

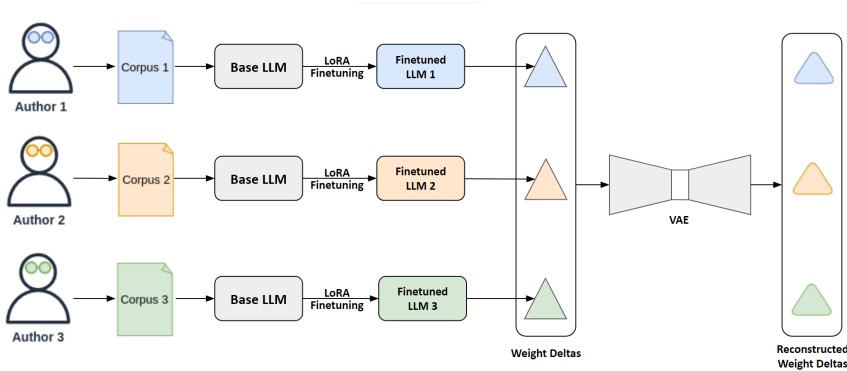

Figure 1: A schematic of how our approach constructs the latent space.

## 3 METHODOLOGY

In this section, we present our proposed approach for few-shot style-conditioned text generation. We frame this as the task of obtaining an LLM finetuned on a given small corpus ($< 10$ samples) of text in a certain style. The finetuned LLM is then capable of generating text in that style. Given a pre-trained autoregressive LLM $P_\Phi(y_t|x, y_{<t})$, where $\Phi$ represents the base model weights, the same LLM finetuned on a corpus $C$ is represented as $P_{\Phi + \Delta\Phi_C}(y_t|x, y_{<t})$ where $\Delta\Phi_C$ stands for the difference in model weights (*weight delta*) on account of fine-tuning the base LLM. As a shorthand, we represent the weight deltas as $\Delta_C$ and the corresponding finetuned LLM as $P_{\Delta_C}$ in the rest of the paper. Thus, our task can be expressed as follows: given a small text corpus $C^*$ from a certain author, we find $\Delta_{C^*}$ that, when applied to the base LLM weights, produces the finetuned LLM $P_{\Delta_{C^*}}$, which generates text that mimics the style of that author. Our overall approach consists of two steps: 1) constructing a latent space of LLM weight deltas and 2) approximating novel finetuned models via interpolation in this latent space. We discuss these steps in the following sections.

### 3.1 LEARNING A LATENT SPACE OF WEIGHT DELTAS

In order to construct a latent space of LLM weight deltas, we extract the deltas and use them to train a VAE. This process is summarized in Figure 1. We start with text corpora belonging to distinct authors exhibiting a developer-chosen number of distinct writing styles. We finetune a base LLM on each corpus using LoRA to obtain our finetuned models. The weight deltas are then extracted from these models and used as inputs to the VAE, which in turn yields a latent space of these deltas.

#### 3.1.1 EXTRACTING WEIGHT DELTAS VIA LoRA

For the pretrained LLM $P_\Phi(y_t|x, y_{<t})$, we extract a collection of model weight deltas $\Delta_1, \Delta_2, \Delta_3, \ldots, \Delta_n$ corresponding to instances of the original LLM finetuned on text corpora $C_1, C_2, C_3, \ldots, C_n$, with each corpus belonging to a distinct author. Each corpus must be large enough for finetuning to capture the style of each text, though we note this limitation does not extend to novel styles encountered after training the latent space. Thus we can apply the approach to authors with far fewer samples than what would normally be needed to finetune the LLM. When applied to the base LLM, the extracted weight deltas produce the finetuned LLMs $P_{\Delta_1}, P_{\Delta_2}, P_{\Delta_3}, \ldots, P_{\Delta_n}$ respectively. This approach makes no assumptions about the finetuning process and thus any fine-tuning method could be applied so long as the differences are captured sufficiently in all or a subset of model weights. For our experiments, each corpus $C_i$ contained at most 300 text samples, with each sample having a maximum of 60 tokens. In order to make the behavior of the system more predictable, we perform finetuning using the same hyperparameters for all authors so that differences between weight deltas are only dependent on the training corpora. To reduce the dimensionality of these weight deltas and the computational cost of finetuning, we capture approximations of the weight deltas by applying LoRA and further reduce the dimensionality via Principal Component Analysis (PCA).

### 3.1.2 EMBEDDING WEIGHT DELTAS VIA VAE

Using these extracted weight deltas as input, we train a variational autoencoder (VAE) (Kingma & Welling, 2013) to learn a low-dimensional latent representation of the extracted deltas. Given an input weight delta $\Delta_C$, the VAE encoder outputs two latent vectors corresponding to the mean $\mu$ and covariance $\sigma$ matrices that define a Gaussian distribution. The VAE decoder samples from this distribution and outputs the reconstructed weight delta $\Delta'_C$. The VAE loss function is given by:

$$Loss = \|\Delta_C - \Delta'_C\|^2 + KL(N(\mu_{\Delta_C}, \sigma_{\Delta_C}), N(0,1)), \tag{1}$$

In our experiments, we found it useful to weigh the KL term using a tunable parameter $\beta$ (Higgins et al., 2017) giving us the loss function:

$$Loss = \|\Delta_C - \Delta'_C\|^2 + \beta \cdot KL(N(\mu_{\Delta_C}, \sigma_{\Delta_C}), N(0,1)). \tag{2}$$

We found that setting $\beta < 1$ helped the VAE to adequately reconstruct the weight deltas due to their small scale and high variability. However, a side effect of weighing the KL term was that this led to the latent space being partially disconnected. We mitigated this by filtering out disconnected data points to extract a continuous subset of the latent space.

Our filtering algorithm starts by sorting the data points based on the sum of their reconstruction and KL losses, in ascending order. This allows differentiating between points that the VAE fit well (low total loss) from those that it did not (high total loss). For each dataset or split of a dataset, we extract $n$ data points with the lowest total loss ($n = 5$ for this paper's experiments), which we then average to find their centroid in the latent space for this dataset or split of the dataset. Then, the maximum of their Euclidean distance from this centroid is calculated. For each data point in the dataset, we calculate the distance between itself and the centroid of its dataset and compare that to the aforementioned maximum Euclidean distance multiplied by a factor. If the distance to the centroid is less than that threshold, it is considered a part of the continuous subset of the latent space. If not, it is discarded. We found that a factor of 3 was capable of adequately distinguishing between points that did and did not belong to the continuous subspace of the latent space for all datasets, verified by inspecting the latent space graphically and using Euclidean distance. These results are shown in the appendix. For the results that follow in the main paper, we only make use of the continuous subset of the latent space obtained via filtering.

### 3.2 GENERATING WEIGHT DELTAS VIA INTERPOLATION

After training the VAE and filtering the latent space, we generate new weight deltas and their associated models via interpolation within the VAE latent space. We perform linear interpolation guided by a random sample from the finetuned models $P_{\Delta_1}, P_{\Delta_2}, P_{\Delta_3}, \ldots, P_{\Delta_n}$. This approach does not rely on any additional hyperparameters or domain knowledge for its operation, making it generalizable to other domains and datasets. To simplify interpolation, this method assumes that the topology of the latent space is smooth, which we ensure with our clustering-based filter algorithm in this work. Given a small corpus $C^*$ containing a few text samples from an unseen author, we pick $K$ models at random from the collection of finetuned models and perform one pass of finetuning, i.e., one-step of backprop, on each using $C^*$. Due to the small size of $C^*$, this step is not computationally intensive. We perform a single step of backprop for each group of samples to reduce the time and resources required to perform inference as well as to avoid overfitting. We found that as we increase the number of text samples and average these changes, our approximation improved. This can be seen as equivalent to doing multiple gradient steps. We compare against one-step and not multiple steps of finetuning on the same samples for a fair comparison. This finetuning yields $\Delta_1^*, \Delta_2^*, \ldots, \Delta_K^*$ which vary only slightly from the original $\Delta_1, \Delta_2, \ldots, \Delta_K$ and are thus incapable of modelling the style of $C^*$. However, using the VAE, we interpolate the weight delta corresponding to $C^*$ using these slight variations. We forward $\Delta_1, \Delta_2, ..., \Delta_K$ as well as $\Delta_1^*, \Delta_2^*, ..., \Delta_K^*$ through the VAE encoder to get $\mu_1, \mu_2, ..., \mu_K$ and $\mu_1^*, \mu_2^*, ..., \mu_K^*$ respectively. We treat each pair $(\mu_t, \mu_t^*)$ as a vector in the latent space moving from the original model $\mu_t$ in the direction $\vec{r}_t = \mu_t^* - \mu_t$. This gives us $K$ latent-space vectors, which, given the saliency and continuity of the latent space, should guide us towards a point that approximates the target model corresponding to $C^*$ when passed through the decoder. For each pair of $N$-dimensional lines $(\vec{\mu_a}, \vec{r}_a)$ and $(\vec{\mu_b}, \vec{r}_b)$, assuming they point towards our target model, we want to find their intersection point in the latent space. Expressed mathematically, we find $t_a$ and $t_b$

that satisfy the following equation:

$$\vec{\mu_a} + t_a\vec{r_a} = \vec{\mu_b} + t_b\vec{r_b} \tag{3}$$

which can be re-written as:

$$\vec{r_a}t_a - \vec{r_b}t_b = \vec{\mu_b} - \vec{\mu_a} \tag{4}$$

or in matrix form:

$$Ax = B, \tag{5}$$

where

$$A = [\vec{r_a} \quad -\vec{r_b}]^T$$
$$B = [\vec{\mu_b} - \vec{\mu_a}]$$
$$x = \begin{bmatrix} t_a \\ t_b \end{bmatrix}$$

Solving this equation returns the vector $\vec{\mu_{C^*}} = \vec{\mu_a} + t_a\vec{r_a} = \vec{\mu_b} + t_b\vec{r_b}$ which, when passed through the decoder, reconstructs our target $\Delta_{C^*}$. However, since there is no guarantee that two $N$-dimensional lines will intersect, we modify this equation to instead find the least-square approximation:

$$\underset{x \in \mathbb{R}^N}{\arg\min} \|Ax = B\|, \tag{6}$$

Solving this equation returns $t_a$ and $t_b$ that represent the closest point on each line to the other one. In this case, we return the midpoint of the line connecting the two points as our target $\vec{\mu_{C^*}}$, which is then forwarded through the decoder to be reconstructed. For certain lines, this point falls in the negative direction of the vector from the source to the target models, indicating that the two models are diverging in one or more latent dimensions. We do not consider the interpolated models to be valid in such cases. For our experiments, we sampled $K = 3$ models to perform interpolation.

Interpolated models differ based on the choice of source models. For our results, for each test data point, we choose base models from a different dataset or different dataset split than that which the test point belongs to. The Reddit dataset being naturally split into subreddits facilitates this. For the Gutenberg and Twitter datasets which do not have such a split, we train the VAE on both for a similar analysis. Results when ignoring this consideration (i.e., testing with base models from the same dataset as the test point) are shown in section A.4.3 of the appendix.

We experimented with two interpolation methods that differ in how they handle successive text samples. For simple interpolation, we perform linear interpolation directly using only the changes in latent space produced in the current timestep. For accumulative interpolation, we accumulate and average the changes in the latent space produced across all previous timesteps until the current one. This helps in stabilizing the interpolated models. This stability issue is discussed in the appendix in section A.4.4. The results produced using accumulative interpolation are discussed in the results section while those obtained using simple interpolation can be found in the appendix in section A.5.

## 4 EXPERIMENTS

### 4.1 DATASETS

We evaluated our approach using three datasets—1) the Twitter dataset, 2) the Gutenberg dataset and 3) the Reddit dataset. The Twitter dataset is a subset of the Sentiment140 dataset which contains tweets tagged with the Twitter handles of their authors (Go, 2009). We filtered out authors with fewer than 200 tweets in order to ensure there were enough text samples to finetune the LLM, ending up with 17 authors in total. The Gutenberg dataset was collected from the Project Gutenberg website (Project Gutenberg) which offers free access to electronic books. We retrieved the top 100 most popular books of all time and treated each book as a separate author with a distinct style. We feel this is a reasonable assumption since an author's style may change from one book to another, especially in fiction. We also made use of the Reddit dataset, a subset of the Reddit Million User Dataset (MUD) (Baumgartner et al., 2020). Similar to Khan et al. (2023), we focused on four subreddits with distinct styles, namely: r/wallstreetbets, r/news, r/AskHistorians and r/australia. We

filtered out authors with fewer than 200 posts and picked 30 at random from each subreddit, ending up with a total of 120 authors.

For each dataset, we set aside 10% of the authors as the test data and used the remaining for training. We split the datasets by author to ensure that the styles of the authors in the test data were not seen by the VAE during training. For both training set and test set authors, we further split each author's text corpus into train, validation and test sets which we used to finetune and evaluate the LLMs.

## 4.2 FINETUNING LLAMA-2 WITH LORAS

We used the open-source autoregressive LLM Llama-2-7b (Touvron et al., 2023) as the base LLM in this work. Since the model has a large number of weights (7 billion), we apply LoRA finetuning instead of finetuning the original parameters by adding an adaptation layer to each Q and V attention layer of the base LLM. LoRA allows us to both save on computational resources as well as directly use the adapter weights as the weight deltas for our system. Finetuning is done via next-token prediction on each corpus using cross-entropy loss. Llama-2-7b contains 32 decoder units, each with four 4096×4096 attention layers. This amounts to a total of more than 2 billion weights. Applying LoRA to the Q and V matrices, each unit now contains four $r$×4096 vectors instead where $r$ refers to the rank of the LoRA adapters. We set $r$ to 2 to decrease the number of weights to about 1 million in this work. Thus, for each author corpus, we finetune an instance of Llama-2 using LoRA to obtain a 32×4×2×4096 weight delta. Additionally, we used an alpha value of 8 and a dropout rate of 0.1. When the LoRA adapter layers are populated by the weight delta, we obtain a finetuned model that generates text in the style of that author.

## 4.3 VAE TRAINING

For the VAE encoder and decoder, we used fully-connected networks as there is no inherent structure to the model weight deltas. The VAE architecture used in this work is shown in Figure 2. We used a latent dimension of size 8 and trained the VAE for 600 epochs at a learning rate of 1e-4, using the Adam optimizer and a $\beta$ value of 0.03. To reduce GPU memory and training time requirements, we applied a compression step to reduce the number of values in the weight deltas. Even though we set the LoRA rank to the smallest possible value for effective finetuning, we found that the dimensionality of the weight deltas still posed a bottleneck on the GPU memory available for training the VAE. We thus compressed the weight deltas using Principal Component Analysis (PCA) due to its simplicity and negligible GPU processing requirements. We trained 32 PCA models (one for each Llama-2 decoder unit) to reduce the dimensionality of the weight deltas from 4×2×4096 to 4×2×$P_d$ where $P_d$ is the number of output components of the PCA model. We found PCA to be capable of reconstructing the model weight deltas, even unseen ones. For the Reddit corpus, setting $P_d = 400$ led to the PCA model explaining 98% of the average variance. The 2% loss corresponded to negligible effects on the performance of reconstructed weight deltas as verified by assessing their generated text and a quantitative assessment of their cross-entropy losses, which we give in the appendix. We thus trained the VAE using these PCA-reduced weight deltas.

## 4.4 COMPUTE RESOURCES

All experiments were conducted using the cloud computing resource provided by $<$ Redacted for Anonymity $>$, consisting of 18 CPU cores and 2xNVIDIA v100l GPUs with 32GB of memory each.

## 4.5 METRICS

We used two metrics for analyzing our experimental results. First, we used the cross-entropy loss on the test split of the corpus of a given unseen author. Using this loss, we compared the performance of the interpolated models to the finetuned source models. Intuitively, this provides a quantitative estimate about the extent to which the interpolated models use the same words in the same order as the reference author corpus. As this metric assumes access to the output probabilities of the LLM, it cannot be used to compare our approach with closed-source models. Since LLMs have a different base cross-entropy loss for different writing styles, to aggregate the losses among different authors, we normalized each author's cross-entropy loss using their base LLM cross-entropy loss.

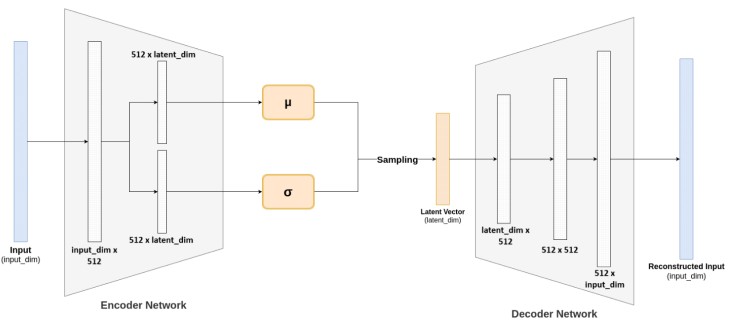

Figure 2: The Variational Autoencoder architecture that we used in our implementation.

The second metric we used was the Universal Authorship Representation (UAR) model, proposed by Rivera-Soto et al. (2021) to generate author style embeddings and measure the cosine similarity between the test split and the text generated by the models being evaluated.

### 4.6 BASELINES

We compared the performance of the interpolated models to the finetuned source models as well as prompting both GPT-3.5 and the base Llama-2 for style-conditioned text generation. We also compared the performance on the Reddit dataset to results reported in the related literature. To the best of our knowledge, StyleMC is the only prior system that targets the problem of few-shot style-conditioned text generation with a number of samples roughly equivalent to our work. Unfortunately, the source code for StyleMC has not yet been made available and thus we cannot reproduce their results. Instead, we directly quote the results as reported by the StyleMC authors.

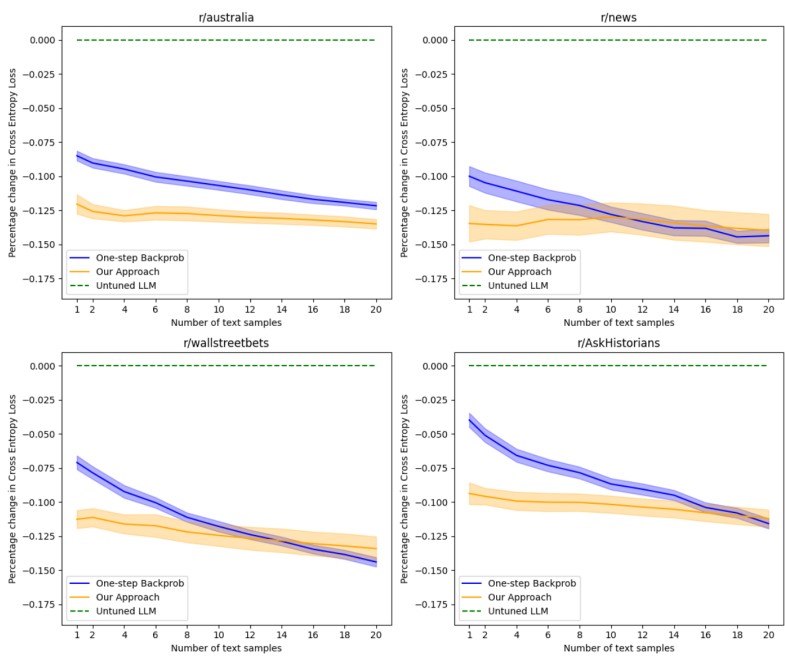

Figure 3: Cross-Entropy Loss Results for the Reddit dataset.

## 5 RESULTS

In this section, we present the results of our experiments, organized by dataset.

| Method | UAR (2 samples) | UAR (16 samples) |
|---|---|---|
| Our Approach | **0.609** | 0.607 |
| One-step Backprop | 0.588 | 0.601 |
| Prompting GPT-3.5 | 0.581 | 0.649 |
| Prompting Llama-2 | 0.592 | 0.633 |
| StyleMC | - | **0.849**† |

Table 1: Comparison of different methods using the UAR similarity metric. The dagger symbol indicates that the values are reported verbatim from their respective sources.

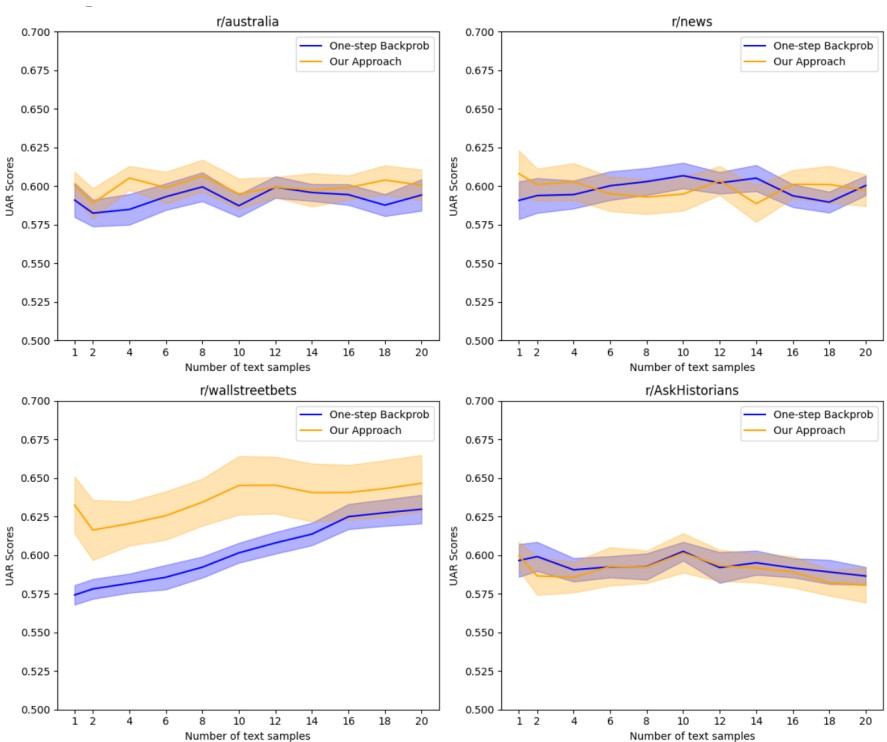

Figure 4: UAR Scores for the Reddit dataset comparing our approach with finetuning.

## 5.1 REDDIT RESULTS

Figure 3 shows the cross-entropy losses obtained for the Reddit dataset, grouped by subreddit. We show the percentage change in the loss, relative to the base cross-entropy loss, on the y-axis. All other losses are below this base loss since finetuned models are better at modeling the text corpus compared to the base model due to similarities among Reddit posts even if they are taken from different subreddits. These results also show that our approach outperforms finetuning for all subreddits for low numbers of text samples ($< 10$). As more text samples become available, the performance of our approach approximately converges to that of finetuning.

Figure 4 shows the cosine similarity scores for the UAR style embeddings. The only subreddit which exhibits a difference between the finetuned models and our approach is *r/wallstreetbets*. For other subreddits, the performance is mostly equivalent.

Finally, we compare our approach with GPT 3.5 prompted to perform style-conditioned text generation. Results are shown in Figure 5. We find that though GPT 3.5 outperforms our approach in some cases, the variance in its performance is much larger and suggests that it is less reliable for this task. We note that GPT 3.5 is also many times larger than our model and may contain the Reddit data in its training corpus.

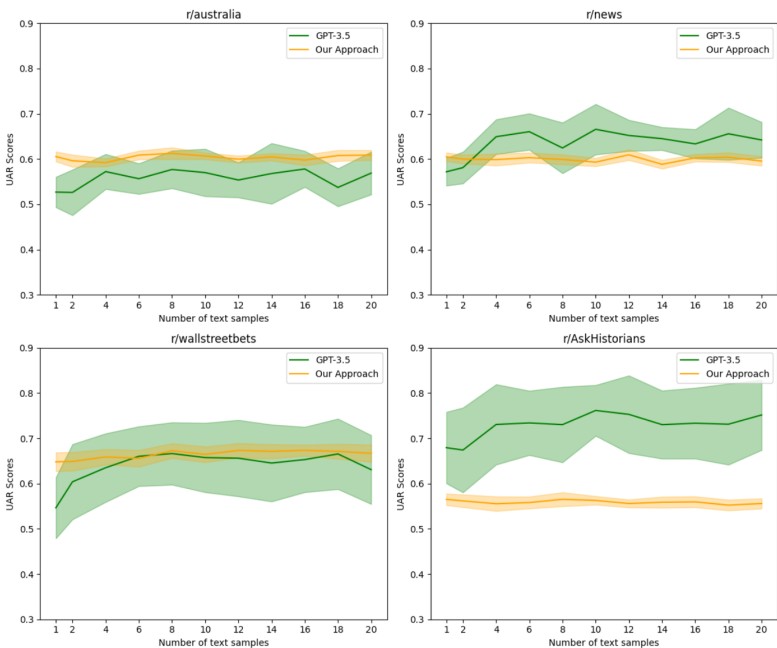

Figure 5: UAR Scores for the Reddit dataset comparing our approach with GPT-3.5.

Table 1 shows the UAR scores for our approach and each baseline, aggregated across all subreddits. These results confirm our prior observations about the performance of our approach compared to finetuning and prompting GPT 3.5. We also include results obtained by prompting an instruction-tuned version of Llama-2 for comparison. We see that the reported performance of StyleMC with 16 samples exceeds the performance of our approach. However the StyleMC authors do not provide any details about the distribution of their performance scores. It is worth noting that they report a higher mean score for GPT 3.5 (0.742) compared to what we obtained (0.649). This may suggest that though we used Reddit data extracted from the same subreddits, our choice of users might differ from those used in evaluating StyleMC, leading to negative skew in our results.

While StyleMC does better when using 16 samples, our approach exhibits roughly the same performance regardless of sample size as shown in Table 1 and Figure 6. The results in this figure also confirm our previous observation that though prompting outperforms our approach as the number of samples increases, our approach has a smaller margin of error and is thus arguably more reliable for this task. We run the Mann-Whitney U Test between these distributions, and find that our approach has a significantly higher distribution (better scores) compared to one-step backprop (finetuning) and prompting GPT-3.5 for the 2 sample case.

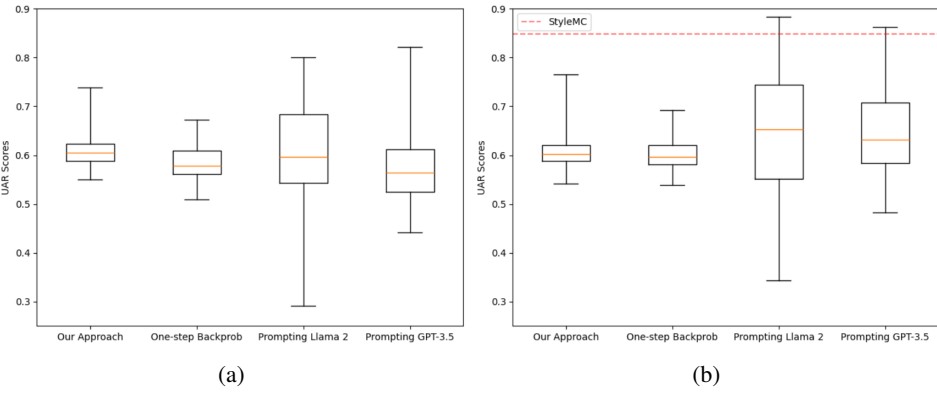

Figure 6: Comparison of different methods using UAR with (a) 2 and (b) 16 text samples.

## 5.2 TWITTER AND GUTENBERG RESULTS

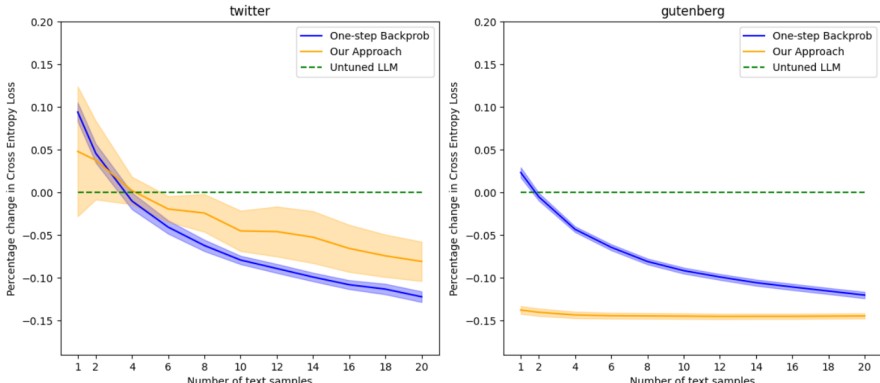

Figure 7: Cross-Entropy Loss Results for the combined Gutenberg and Twitter datasets.

Figure 7 shows the cross-entropy loss results for the Twitter and Gutenberg datasets. As before, the percentage change in cross-entropy loss is shown on the y-axis. The results for Gutenberg are similar to those for Reddit with our approach outperforming finetuning for low numbers of samples and converging to finetuning as the number of samples is increased. However, for Twitter, we find that finetuning slightly outperforms our approach, particularly as the number of text samples is increased. We suspect this may be due to the Twitter dataset's smaller size compared to the Gutenberg dataset. Thus the VAE is unable to learn model representations that are salient enough for useful interpolation.

We computed UAR scores for the Gutenberg and Twitter datasets as well. However, these metrics failed as the UAR representations are trained on Reddit data and so are inadequate for representing out-of-distribution text styles. We report and discuss these results in the appendix in section A.7.

## 6 CONCLUSION

We presented an approach for style-conditioned text generation using LLMs in a few-shot setting. We demonstrated that a VAE is able to encode meaningful style information in the form of latent embeddings of LLM weight deltas (finetuned via LoRA). We also presented evidence that interpolation in this latent space enables generating new model weights that outperform finetuning and other baselines in low-resource ($n < 10$) settings.

## 7 ETHICS STATEMENT

We acknowledge the potential harm of dual use, or use by bad actors, from this research. Few-shot style-conditioned LLM text generation expands the possible applications of LLM technologies. But these possible applications include bad actors using this to mimic a target's writing style for impersonation or conditioning an LLM on their own style of writing to mask their use of LLM generated text. We plan to acknowledge this risk in the public release of the code. But we do identify that both of these applications are already possible, and more work is needed in detecting and combating such uses.

## 8 REPRODUCIBILITY STATEMENT

For reproducibility, we make use of only publicly available datasets. We also make use of an open source LLM for all experiments. Finally, we include all code in the supplementary materials.

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

# A APPENDIX

## A.1 PCA-REDUCED WEIGHT DELTAS

As discussed in section 4.3, the difference in the percentage change in cross-entropy loss for the original and PCA reconstructed weight deltas is shown in Figure 8.

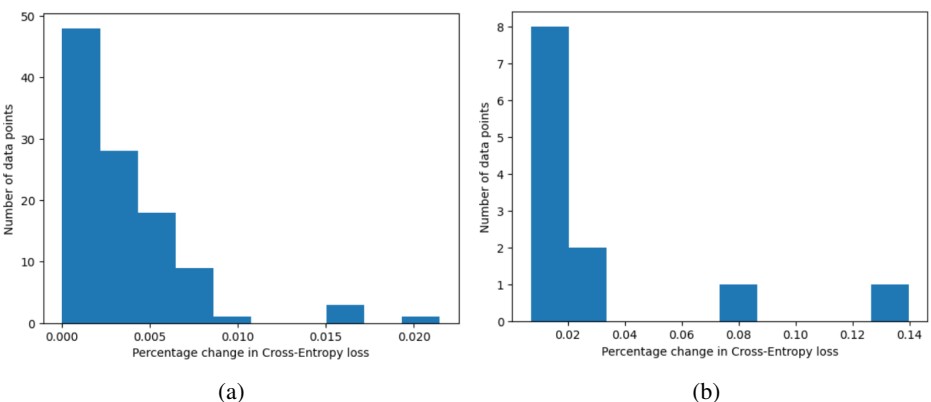

(a)                                              (b)

Figure 8: Histograms of the difference in percentage change in cross-entropy loss between the original and the PCA reconstructed weight deltas for (a) the train split and (b) the validation split of the Reddit dataset.

## A.2 REDDIT DATASET LATENT SPACE

The latent space of the VAE trained on the Reddit dataset is shown in Figure 9a. As can be seen, the latent space is fragmented and discontinuous. However, we were able to extract a continuous subset of the latent space via the filtering algorithm as described in section 3.1.2, consisting of 40 data points, constituting 37% of the training set. This filtered subset of the latent space is shown in Figure 9b. We see here that in this case, the VAE is able to differentiate between the weight deltas that belong to the different subreddits without any prior information.

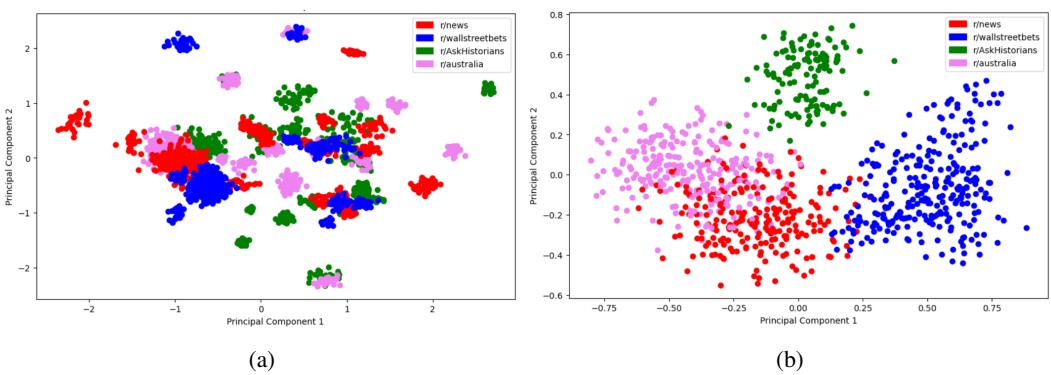

(a)                                              (b)

Figure 9: Two-dimensional projection of the VAE latent space trained on the Reddit dataset for (a) the full latent space, and (b) the connected subset of the latent space.

We confirm this observation by computing the Euclidean distance in the latent space between data points based on the subreddits they belong to. The heatmap in Figure 10b confirms that in this filtered latent subset, data points from the same subreddit are closer to each other than to data points from other subreddits. This is in contrast to the heatmap in Figure 10a which shows that this does not apply to the full, unfiltered latent space.

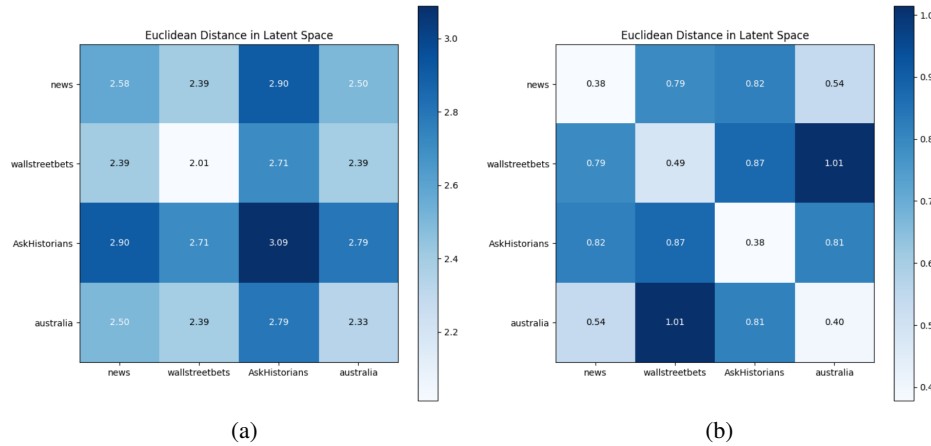

(a)  (b)

Figure 10: Euclidean distance between subreddit weights in the VAE latent space for (a) all data points, and (b) data points belonging to the continuous subset of the latent space.

## A.3 TWITTER AND GUTENBERG LATENT SPACE

As with the Reddit data, we found that the full latent space of the VAE trained on the combined Twitter and Gutenberg dataset was disconnected. However, as shown in Figure 11, in this case, a larger portion of the data fit within the continuous subset of the latent space. 11 points and 66 points were encoded in this continuous subset for Twitter and Gutenberg respectively, accounting for 78% and 81% of their respective training data. We hypothesize that the higher percentages are due to higher inner similarity among data points within these datasets, compared to Reddit. Again, the VAE is able to discern the differences between both datasets from the weight deltas alone.

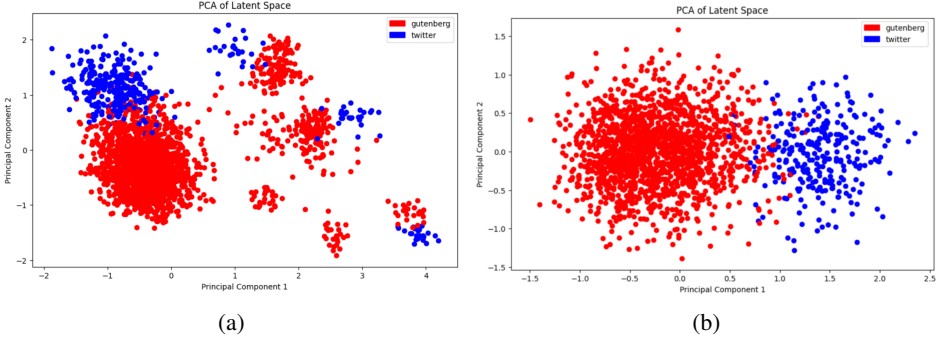

(a)  (b)

Figure 11: Two-dimensional projection of the VAE latent space trained on the combined Twitter and Gutenberg datasets for (a) the full latent space, and (b) the connected subset of the latent space.

## A.4 INTERPOLATION CASE STUDY

In this section, we present a case study showcasing the functionality of our interpolation process. We pick a data point from the training set in order to compare its interpolated latent representation with its actual latent representation. First, we show a sample result where our approach successfully interpolates latent models. Second, we study how the choice of source models and data samples affect the results. Finally, we show how accumulative interpolation solves some of the issues.

### A.4.1 SUCCESS SAMPLE

We picked a datapoint from the subreddit r/wallstreetbets. We randomly select source models belonging to other subreddits. Here we pick 3 base models. Figure 12 shows the cross-entropy results produced by models interpolated using our approach compared to finetuning. As can be seen, our

approach is able to interpolate models that outperform finetuning on the given samples. Given the actual position of the datapoint in latent space, we can inspect the operation of our approach, as shown in Figure 13. We also validate these results in Figure 14 by plotting the Euclidean distance in latent space between the interpolated models and the actual location of the target model. These figures show that our approach indeed captures the style of the target author as the number of available text samples increases.

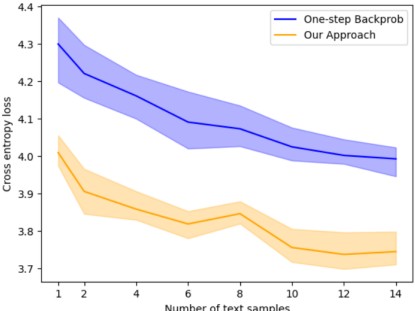

Figure 12: Cross-Entropy Loss Results for the case study data point.

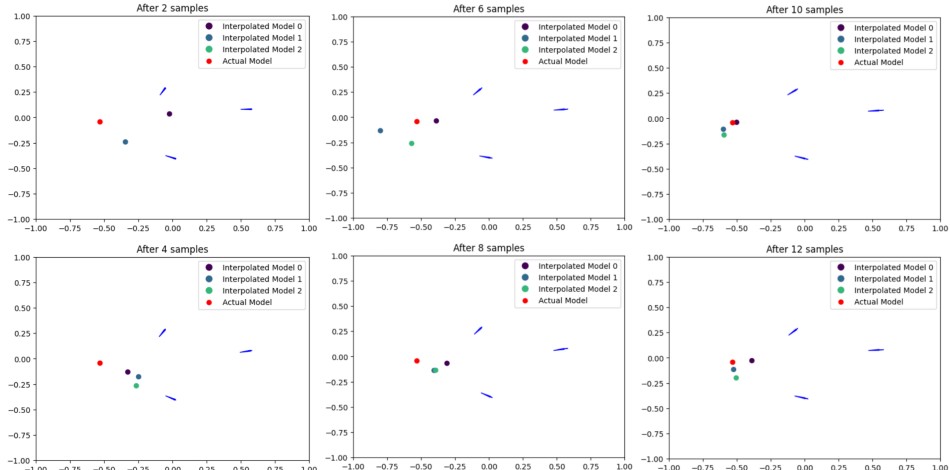

Figure 13: A sequence of snapshots from the latent space during the operation of our system. The red dot represents the actual location of the target model in the latent space. The interpolated models successively approach the actual location as the number of samples increase.

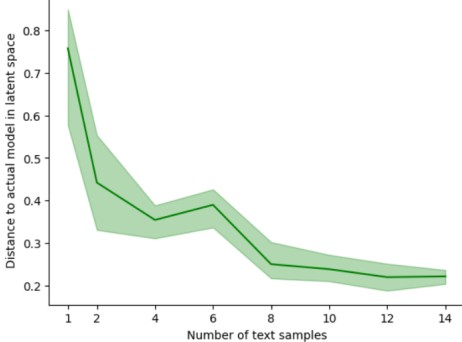

Figure 14: Euclidean distance between interpolated models and the actual model in latent space.

### A.4.2 Effect of Text Samples

While a certain author's text corpus can be viewed as a general representation of their style overall, each sample within the corpus can differ in how much it represents that style. Thus, in few-shot settings, there is a risk that a few text samples are not clearly representative of the author and may misguide the system. Figure 15 shows the cross-entropy results for our approach in the same setting but with a different subset of text samples. We can see that the text samples in the range from 6 to 10 push the system away from the target model. This is supported by observing the latent space in Figure 16. Hence, it is crucial that text samples in few-shot settings are chosen to be clean and representative of the target style. We address this issue by performing accumulative interpolation, as discussed in the section A.4.4.

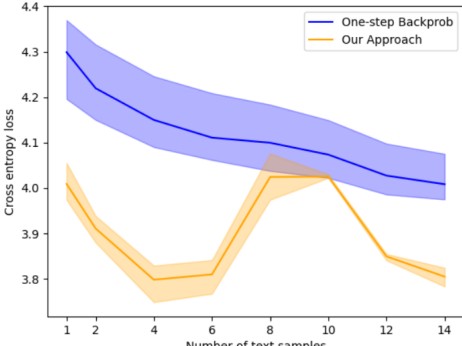

Figure 15: Cross-Entropy Loss Results for different text samples of the case study data point.

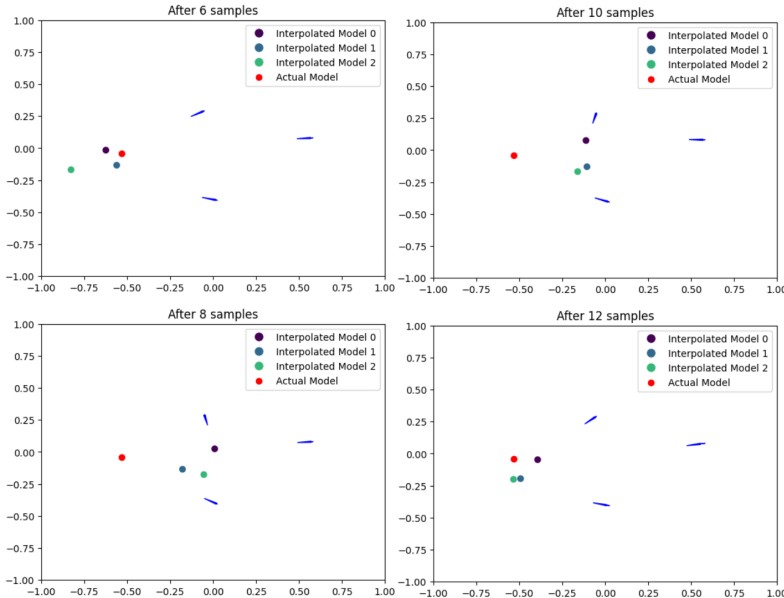

Figure 16: A sequence of snapshots from the latent space during the operation of our system with a bad choice of text samples.

### A.4.3 Effect of Source Models

Since our approach relies on changes in the source models during finetuning to interpolate finetuned models in the latent space, the choice of source models can have a large impact. In Figure 17, we show the cross-entropy results for source models that belong to the same subreddit as the target model (r/wallstreetbets). Two observations can be made here: (1) the source models perform much

better than source models that did not belong to the same subreddit, and (2) the performance of the interpolated models is not as stable as before. Observing the latent space in Figure 18, we find that due to the proximity of the base models to the target model in the latent space, interpolation becomes more difficult due to the instability of the direction of changes in the latent space. We discuss one possible solution for this issue in the next section

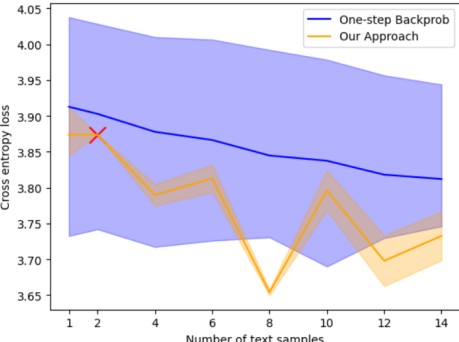

Figure 17: Cross-Entropy Loss Results for the case study data point with source models that belong to the same subreddit. The red X represents failure in interpolation.

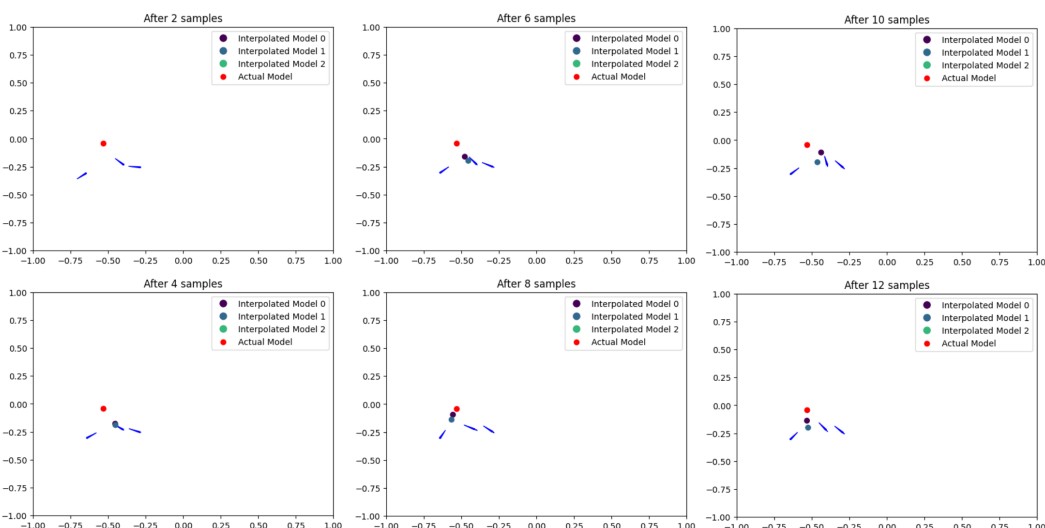

Figure 18: A sequence of snapshots from the latent space during the operation of our system with source models that belong to the same subreddit as the target model.

### A.4.4  EFFECT OF ACCUMULATIVE INTERPOLATION

Through observing the latent space, we found that poor choice of text samples or source models can lead to unstable results. To address this, we proposed accumulative interpolation as described in section 3.2. Figure 19 shows its effectiveness in mitigating stability issues for various cases.

### A.5  SIMPLE INTERPOLATION USING THE REDDIT DATASET

We had previously only shown results of applying accumulative interpolation using the Reddit dataset. Figure 20 shows the cross-entropy results for the Reddit dataset with simple interpolation. Compared to Figure 3 in section 5.1, we find that simple interpolation leads to more visible variations in the performance of the interpolated models.

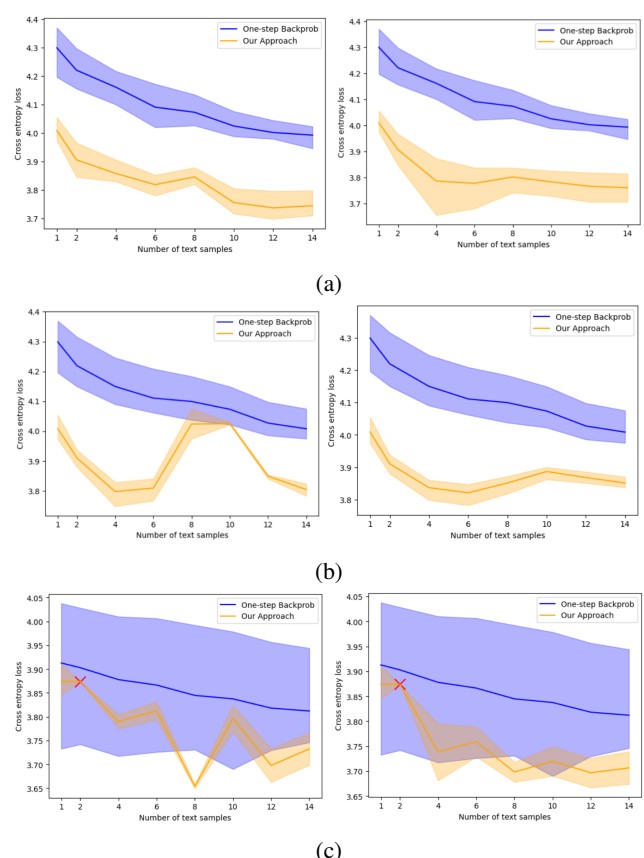

(a)

(b)

(c)

Figure 19: Cross-entropy results using simple interpolation (left) and accumulative interpolation (right) for (a) a good choice of source models and text samples, (b) the same source models but with different text samples, and (c) source models that belong to the same subreddit.

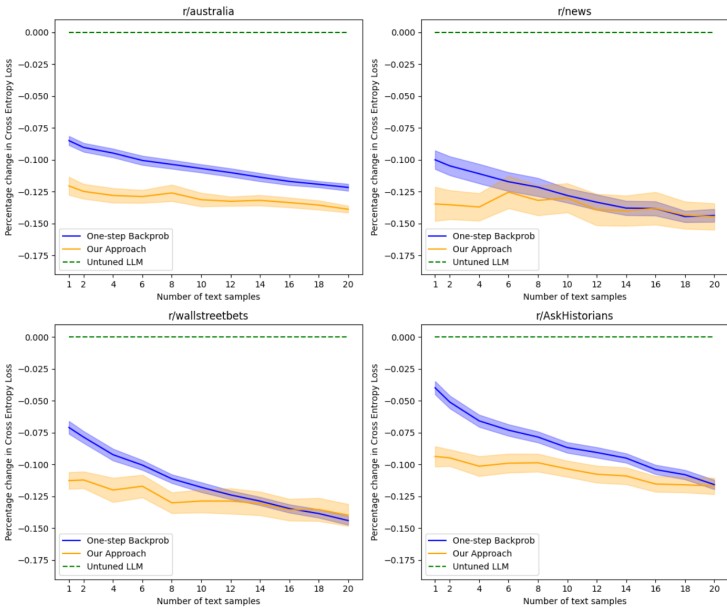

Figure 20: Cross-Entropy Loss Results for Reddit Dataset with Simple Interpolation.

### A.6 REDDIT RESULTS WITH BASE MODELS FROM SAME SUBREDDIT

Figure 21 shows the cross-entropy results of the Reddit dataset when base models are selected to be belonging to the same subreddit as the target model. To compare with Figure 3, we show the results for accumulative interpolation. We find that when the base models already belong to the same distribution as the target model, the performance of our approach is almost the same as finetuning.

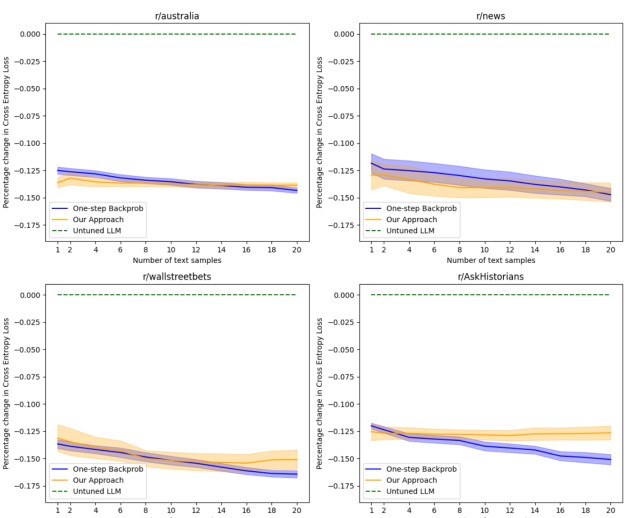

Figure 21: Cross-Entropy Loss for Reddit data with base models that belong to the same subreddit.

### A.7 TWITTER AND GUTENBERG DATASET UAR RESULTS

We report UAR results for the Twitter and Gutenberg data in Figure 22. For Gutenberg, we find that source models that do not belong to the dataset (Figure 22a) perform better on average than those that do (Figure 22b). This is probably caused by UAR being trained on Reddit data that is significantly out-of-distribution for the Gutenberg data. This hypothesis is supported by the relative consistency of the UAR scores for the Twitter dataset which is closer to the style of Reddit data.

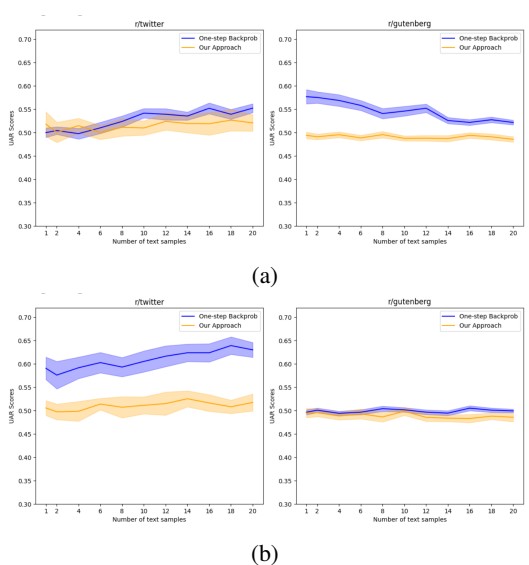

Figure 22: UAR for combined Twitter and Gutenberg data with base models belonging to (a) the other and (b) same dataset.

## A.8 GPT-2 EXPERIMENT

To investigate the generality of our approach, we present further results applying our approach with GPT-2 (Radford et al.) as the pretrained model instead of Llama-2 as in the previously discussed experiments. Figure 23 shows the latent space of a VAE trained on the weight deltas of finetuned GPT-2 models using the combined Twitter and Gutenberg dataset. As in experiments with Llama-2 (Figure 11b), the VAE is again able to differentiate between Twitter and Gutenberg using the weight deltas. We note that for GPT-2, the resulting VAE latent space was not disconnected as in the case with Llama-2, and thus we did not have to run the filtering algorithm to obtain a connected latent space. We hypothesize that this may be due to the fact that we finetuned GPT-2 directly without using LoRA and did not make use of PCA to reduce the weight deltas (due to GPT-2's smaller size). We hope to study this further in future work.

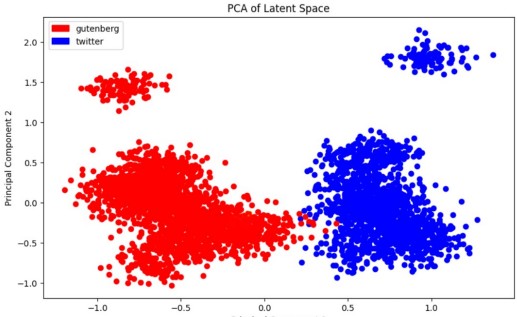

Figure 23: Two-dimensional projection of the VAE latent space on the combined Twitter and Gutenberg datasets using GPT-2 as the base pretrained model for finetuning.

## A.9 STYLE EMBEDDINGS

In this section, we present results evaluating our approach using the style embeddings developed by Wegmann et al. (2022). Similar to the experiments using UAR, we compute the cosine similarity between the style embeddings for the test split and the text generated by our models. Tables 2 and 3 show the scores for our approach and baselines for the Reddit and Twitter/Gutenberg data respectively.

| Method | SE (2 samples) | SE (16 samples) |
|---|---|---|
| Our Approach | 0.523 | 0.558 |
| One-step Backprop | 0.539 | 0.582 |
| Prompting GPT-3.5 | 0.581 | **0.65** |
| Prompting Llama-2 | **0.592** | 0.633 |

Table 2: Comparison of different methods using the style embedding similarity metric (higher is better) on the Reddit dataset.

| Method | SE (2 samples) | SE (16 samples) |
|---|---|---|
| Our Approach | **0.359** | **0.367** |
| One-step Backprop | 0.318 | 0.365 |

Table 3: Comparison of different methods using the style embedding similarity metric (higher is better) on the Twitter and Gutenberg dataset.

These results show that our approach does worse than one-step backprop on Reddit data but better than one-step backprop on the Twitter and Gutenberg dataset. This is contrary to expectation as, similar to UAR, these style embeddings Wegmann et al. (2022) were trained using data from Reddit (Baumgartner et al., 2020). We suspect that this may be due to the two types of embeddings being trained using different subreddits and/or train/test splits of the same overall dataset, and thus,

it is possible that the text used to train the style embeddings were more like the text in our Twitter/Gutenberg dataset than in the specific subset of subreddits in our Reddit data. We intend to study the effect of different style embeddings more thoroughly in the future. Figures 24 and 25 show the cosine similarity scores for the style embeddings for each Reddit subreddit and the Twitter and Gutenberg datasets separately.

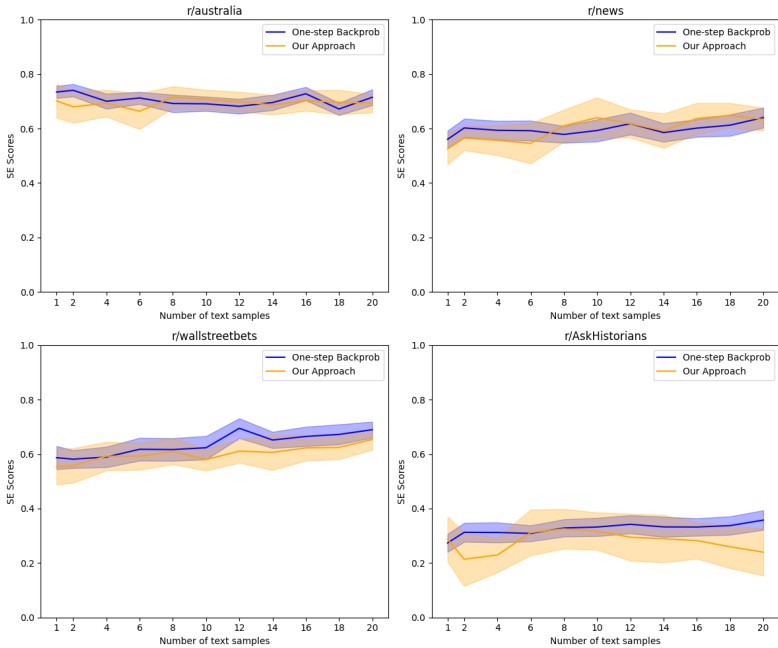

Figure 24: Style embedding scores for the Reddit subreddits comparing our approach with finetuning.

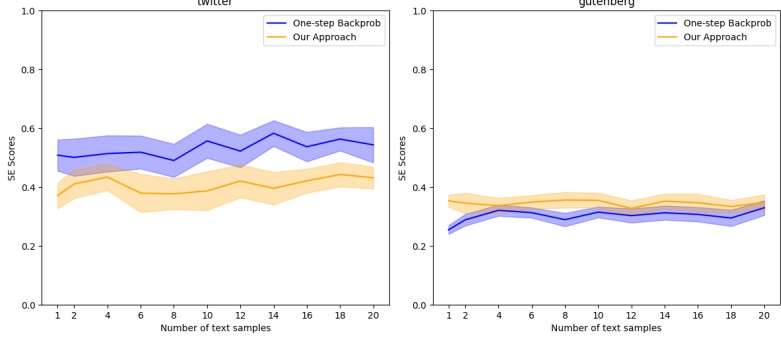

Figure 25: Style embedding scores for the Twitter and Gutenberg dataset comparing our approach with finetuning.

## A.10 SAMPLE GENERATIONS

In this section, we present some sample generations for both the Reddit and the Twitter-Gutenberg datasets in Tables 4 and 5 respectively. In both tables, each row shows a randomly chosen text sample from the training set and randomly chosen samples from our approach with 2 and 16 samples. Samples in a given row correspond to the same corpus (or user in the case of Reddit and Twitter).

| Training Sample | 2 samples | 16 samples |
|---|---|---|
| r\AskHistory | | |
| You're exactly correct. The army only had significant issues with recruitment during times of natural disasters (like the Antonine Plauge or the Plauge of Justinian) or following military emergencies (like Adrianople), as the immediate needs for large volumes of manpower strained demographic resources. | The first time I read the book, I didn't get it at all. hopefully you will like it better the second time. I have a feeling that the sequel will be just as good as the original. I've read the first book twice, and have the sequ | By the time I was old enough to read The Hobbit, my father had been reading the book to me for years. He knew a bit more about Tolkien's life than I do, but I have always wondered whether the fact that he was born in 189 |
| That is a really good point on using this as Hamlet's characterization. But back on the history topic, what's this "Grand Tour of Europe" of Renaissance Crown Princes? | This week is the 10th anniversary of the 2007–2008 financial crisis. This crisis started in the United States, but quickly spread throughout the world. It was not the first financial crisis that occurred in the world, but it is the most famous | You can find a lot of information on this subject on this website, as well as many other sources, including other websites. Here is an example of a website that provides information on the history of this subject: `http://www.history.org/history/teach/themes/amer` |
| r\Australia | | |
| Basically you're ok with corporate interests deciding who can eat, and do all kinds of things. Hmm. Do you know what that kind of system of government might be called? | The most important thing to keep in mind when looking at the polls is that a lot of these polls are self-reporting. I think a lot of people are over-reacting to this. They're not going to be able to stop this, it's going to | Well, it's not a big deal. It was just a joke. The only reason I made that comment in the first place was to try and make a point about how people are so quick to point fingers. I'm not actually offended. Just disappointed in the people |
| You have no idea how popular Home and Away is in Ireland and the UK. Literally everyone watches it, not just teenage girls. | They have no idea. they are a joke and have no idea how to manage the situation. If you were a citizen of the country, they wouldn't be "their" people. You're an a———-, and you don't get to decide whether | The only way to stop this is to stop it at the border. Even then, I don't want to go that far. If you are going to let people in, make them assimilate. They need to learn English and they need to accept our laws. It shouldn't |
| r\News | | |

| | | |
|---|---|---|
| You have a very narrow worldview if you believe everyone that disagrees with you is an idiot. | I think it's just that he has no intention of actually being an MP and he's just going to be another of the "me too" politicians who are going to be around for a decade or two after they've done absolutely nothing. | The first thing I notice is that the new one is a little more aggressive in the front. It looks more like a real car than the current one. I was actually thinking this looks kinda like the current model. The front end looks a little more aggressive, but |
| That is also abhorrent. Seriously dude, I would never support the torture of anyone, for any reason: | I've done both. I think there are some great places for both. It just depends on what you are looking for in your next step. I love living in a big city. I would have never left NY if I had stayed on my original plan to go to law school and become | It is a shame that you are being ridiculed for your belief. However I would encourage you to think about the possibility that your belief may be wrong. It's okay to be wrong. You are allowed to believe in something even though it is wrong, but you have to |
| r\wallstreetbets | | |
| You really dont know what youre talking about. Fundamental is anything looking at the intrinsic value, technical does not. Its a boolean, either you are or you arent, there is no other option. | I'm glad that I'm not the only one who gets a bit frustrated with the "give me a number" question. geprüft.com is the best tool for this I've found so far. It's not the only tool, but I'd | The problem is that if you don't get that 30% in a single year, you're probably in the wrong business. And 30% is a big if. It's not impossible. I don't think it's unrealistic, but I |
| A month from now, they'll have the emergency meeting to put it another month out because f— everyone else | I'm not sure I understand the question, but you should be able to find the answer in the documentation of the package. everybody here has a different definition of "reliable". Some people have very low tolerance for risk and would not be satisfied with a 1% drop over | The only reason I have any faith in it is because I think the market is going to crash, and that it will be a great time to buy. At least I hope so. I'm not sure I'm smart enough to trade options. |

Table 4: Sample generations from models output by our method using the Reddit dataset. Each row corresponds to a different user and depicts: an original text sample from that user, random sample generated by a model output from our approach based on 2 samples from that user and a random sample generated by a model output from our approach based on on 16 samples from that user.

| Training sample | 2 Samples | 16 Samples |
|---|---|---|
| **Anna Karenina** | | |
| relations with him, which she had hitherto avoided thinking about. | that the man had a large hand and long, thin fingers. To this I could give no other reason than the mere fact of my having seen | It was at the door of the little hut that the boy's face first became intelligible to me. I could not see the eyes |
| **Emma** | | |
| not been thinking of him, she had forgotten to think of him—but she was very glad to see him. Mrs. Weston would be at ease. The black mare was blameless; | a time he would have been glad of the company of his wife. But now, he felt, he was too much of a man to be | that I could find no other excuse for my conduct than that I was not a man of principle, but a man of the world. I |
| **Middlemarch** | | |
| I have in my hand is an ingenious contrivance—a sort of practical rebus, I may call it: here, you see, it looks like an elegant heart-shaped box, portable—for the pocket; | the door to me, when I was not in the way. They would even take me out to walk, and let me see the world. | I am sure, if the truth were known, that the good man had an interest in my success; but I had never seen him so deeply |
| **Peter Pan** | | |
| After all, he was the only man that the Sea-Cook had feared. But Peter had no sinking, he had one feeling only, gladness; and he gnashed his pretty teeth with | his way to a little house, at the end of a small street. nobody was there to greet him, nor any one else. but | the end of the week, and he was still a prisoner. His wife, who had a little money, had come to the end of her |
| **Sense and Sensibility** | | |
| her less wretched. | that had fallen upon his shoulder. He lifted it off, and the light of day streamed in upon the room. There was a little | his hand, as he spoke, and a great deal more that was still less complimentary. He then said, "If it is true |
| **Wuthering Heights** | | |
| insulted so shockingly by a stranger! She did not comprehend it; and hard work I had to obtain a promise that she would not lay the grievance before her father. | the most beautiful girl he had ever seen. He was not in love with her, but he had fallen a victim to that species of love | and said that he had been a good deal with the family, and was a particular friend of the lady's father, and had been much |
| **Twitter User** | | |
| How are you this mornoon? | hopefully they will come out soon. This is really a fantastic and valuable post. Thank you for | the most interesting and instructive book I ever read. |
| **Twitter User** | | |
| Just havin a chilled Sunday then? I love Sundays for that reason..you can just sit about an be a lazy b and no one cares. | I am a good fellow, a good fellow, I know, I know. I have not a bad heart. I am not bad, no | his mouth. obviously he was a man of no great culture. But I think he must have had a good deal of practical wisdom, and in |

Table 5: Sample generations from models output from our approach using the Twitter and Gutenberg dataset.