# OpenReview forum: "Few-shot Style-Conditioned LLM Text Generation via Latent Interpolation"
_ICLR.cc/2025/Conference — Submitted to ICLR 2025_

### Official Review · Reviewer_mBaA · 2024-11-01

**Soundness:** 2
**Presentation:** 2
**Contribution:** 2
**Rating:** 3
**Confidence:** 4

**Summary:**

The paper proposes to model agnostic approach to tailor LLM to arbitrary styles. To enable learning from smaller samples, the model first trains a VAE framework that learns a representation for "weight deltas" of the LLM when fine-tuned on a large corpus of stylized text. This lower dimension VAE allows the model to learn a similar representation with a smaller corpus. To that end, for a new style, "k" fine-tuned models are taken at random, a single shot fine tuning is performed and interpolated in the VAE space to arrive at the specific deltas for the given data. Quantitative evaluations illustrate the promise of the approach.

**Strengths:**

The formulation is very neat and well established.
The paper is solving a key problem that is very relevant and valuable

**Weaknesses:**

The paper needs to be tested out well. While there is a lot of potential technical contribution, the paper lacks clear validation of its approach. Given the qualitative nature of the style task, I would have liked to see sample generations and analysis on where the approach is working and where it is not. Does the choice of initial K-finetuned models have an impact on the model? These aspects are not addressed, which when addressed will make this a very strong paper.

**Questions:**

See above

---

> ### Author Response · Authors · 2024-11-21
>
> Thank you for your insightful comments and feedback.
>
> Regarding the choice of K finetuned models, we tested the effect of changing the pool of models to sample the K finetuned models from. This result can be found in the section 'Effect of Source Models' in the appendix (section A.4.3). We will add some text summarizing these results in the main part of the paper.
>
> We have more results that we'll add closer to the end of the discussion period.

---

### Official Review · Reviewer_CGYD · 2024-11-04

**Soundness:** 2
**Presentation:** 2
**Contribution:** 3
**Rating:** 3
**Confidence:** 3

**Summary:**

The paper proposes a novel approach for few-shot style-conditioned text generation. The key innovation is representing writing style through model weight differences rather than predefined features. The approach is evaluated against prompting and traditional fine-tuning across three datasets, showing superior performance in low-resource settings with less than 10 samples.

**Strengths:**

- The idea of representing text style via model weight differences rather than fine-tuning and prompting requires less data and manual effort. Using LoRA and PCA for dimensionality reduction makes the approach more efficient and practical.
- With empirical validation across multiple datasets and case analyses, the proposed method displays clear performance advantages in low-resource scenarios against strong baselines.

**Weaknesses:**

- The VAE latent space filtering approach appears ad-hoc, lacking theoretical justification for the chosen filtering criteria and thresholds.
- The main experimental results are too limited in scope. The authors should reimplement baseline methods to ensure fair comparison across consistent settings,
- Performance varies significantly between datasets (notably poorer on Twitter), which needs more detailed analysis. Besides, it could be evaluated on more challenging cross-domain style transfer tasks [1].
- Missing ablations on key components like VAE architecture choices and latent dimension size.
- Paper writing and figure plotting need careful review:
  - Figure texts are too small. Captions could be more detailed
  - It is better not to maintain present tense throughout, especially when describing methodology and experimental settings.
  - Section 4.2 the notation for weight shapes should use proper mathematical multiplication symbol $\times$ instead of alphabet "x" when specifying dimensions.

[1] StylePTB: A Compositional Benchmark for Fine-grained Controllable Text Style Transfer

**Questions:**

- The proposed method is only tested on llama2 7B, could it be generalized to other architectures?
- Some possible related works on controlled text generation via latent interpolation for reference [2-3].

[2] Deep Extrapolation for Attribute-Enhanced Generation

[3] Extracting Latent Steering Vectors from Pretrained Language Models.

---

> ### Author Response · Authors · 2024-11-21
>
> Thank you for your insightful comments and feedback.
>
> Related works: Thank you for bringing these works to our attention. We have added [3] to the related work section. As mentioned in a response to a prior reviewer, we consider style transfer to be out-of-scope for this specific work and so do not evaluate using the StylePTB benchmark as it focuses on paired style transfer tasks.
>
> Baselines: Regarding the comment regarding consistent settings for baselines, if the paper was interpreted to indicate that the baselines were setting-specific, we want to confirm that this is not the case.
>
> Figures/text: We will fix the captions in the figures, the notation for weight shapes and improve the writing in general. Thank you for pointing these out.
>
> We have more results related to the other questions that we'll add closer to the end of the discussion period.

---

### Official Review · Reviewer_qmsk · 2024-11-04

**Soundness:** 3
**Presentation:** 2
**Contribution:** 3
**Rating:** 5
**Confidence:** 4

**Summary:**

The authors propose a novel method for controlled style generation via mixtures of low-rank adapters and a learned latent space. The authors hypothesize that the most ideal solution would be to fine-tune an LLM for each desired style, however this is cost/time prohibitive. Instead they propose learning these adapters based on a small collection of fine-tuned ‘gold’ adapters. The results indicate their interpolation scheme, enabled by a learned VAE over adapter weights, outperforms several baselines, especially in the very few shot case.

**Strengths:**

* Few-shot style modeling is a difficult problem, with many existing methods falling short of leveraging the full capabilities of LLMs for this task. The authors propose a novel procedure which leverages mixtures of adapters, the end result is an arbitrary adapter matrix which can be applied to the base LLM.
* Existing control methods rely on conditioning on an pre-trained style representation (consider citing https://arxiv.org/abs/2406.15586), this approach using LLM representations themselves to encapsulate style and demonstrates that interpolations between these representations produces an effective way to control style in the few-shot setting.

**Weaknesses:**

* I believe the variables in EQ 3-5 are overloaded? Line 201 refers to \mu_1, \mu_2…as the latent representations of the weight deltas. But EQ 3-5 refers to the same vector as ‘any pair’. If this is interpreted correctly, switching the 2nd set of subscripts to ‘a’ and ‘b’ would probably make it easier to interpret.
* I don’t think Figure 2 adds much value here, I believe the space and description in lines 285-290 can be used more effectively to describe your training procedure for the adapters (i.e. training loss, special tokens if any, preprocessing). A citation of the LoRA paper and hyperparameters used should be otherwise sufficient.
* Table 1: consider bolding the best results in the table.
* Consider adding another automatic style evaluation in addition to UAR: https://huggingface.co/AnnaWegmann/Style-Embedding
* The result discussion is lacking, while the results are written out, further explanation of model shortcomings are necessary.
The authors rely on two metrics: change in cross entropy loss, and UAR similarity. While these may correlate with style control performance to a degree, a human evaluation on the outputs of these systems would be far more convincing.

**Questions:**

* Section 3.1.1 explains how a closed set of adapters are derived based off N authors and their writing samples, details are lacking on how exactly these adapters are fine-tuned, i.e. is it simply next token prediction on the Corpus C_i?
* How many tokens are available in each corpus C_i? There is a claim in this section that suggests that far fewer samples are needed to target authors at test time, but no details on the scale.
* Section 3.2 says that a single gradient update is applied to the pool of adapters based on the target style corpus, and that this is a sufficient \delta for the latent space to decode a target weight matrix. Why is a single step chosen? Is there significant improvement in the predicted weights if N training steps are applied?
* Figure 3 suggests that the proposed method is better in the very few-shot case, and the one-step back prop does better with more samples. Why are we limited to 1 gradient update? How do these results scale with more updates?
* In Table 1, Prompting Llama2 appears to perform significantly better than the proposed approach in the 16 sample case. This requires the same inference compute as the proposed approach, why should the more complex mixture procedure be considered over simply prompting here?
* How necessary is the VAE step? I.e. does simply running a linear combination of the LoRA weight matrices to create an arbitrary one result in a meaningful style output?
* Can your method be used for style transfer in addition to style control? This requires semantic retention in addition to style conditional generation.

---

> ### Author Response · Authors · 2024-11-21
>
> Thank you for your insightful comments and feedback. We address your questions and concerns below:
>
> LoRA finetuning details: You are correct in that finetuning was done by next token prediction on each corpus using cross-entropy loss. Additionally, for LoRA hyperparameters, we used a rank of 2, an alpha value of 8 and a dropout rate of 0.1. We will add these details to the paper along with info about the size of the corpora. Also, as suggested, we will remove Figure 2 depicting LoRA finetuning and replace it with a figure of the VAE architecture as requested by another reviewer.
>
>
> Single gradient update: We did a single step for each group of samples so that the inference would not require a lot of time and resources and also to avoid overfitting on these few text samples. We found that as we increase the number of text samples and average these changes, we get better models (which is shown in the cross entropy plots going down as the number of samples increases). This can be seen as doing multiple gradient steps. We compare against one-step and not multiple steps of finetuning on the same samples so as to have a fair comparison with our method which only uses one step. We'll add these details to the paper.
>
>
> Prompting Llama-2: While Llama-2 does do better on 16 samples and above, we focus on the <16 case and suggest swapping to another approach with increasing sample sizes. We note that as depicted in Table 1, our approach performs best in the very few shot case of 2 samples.
>
>
> Style Transfer: While related to stylized text generation, we consider style transfer to be out of scope for this work. Future work could look to adapt this method to perform style transfer and, as pointed out in the review, this would require retaining the semantic information of the source sentence in addition to style conditioning. We will mention this in the discussion section of the paper and also clarify more explicitly that we don't focus on style transfer in the introductory sections of the paper.
>
>
> Equations/Tables: Thank you for your suggestions regarding the overloaded variables in the equations and bolding values in the table. We will make the necessary changes in the final version of the paper.
>
>
> We have more results related to the other questions that we'll add closer to the end of the discussion period.

---

### Official Review · Reviewer_nsiM · 2024-11-04

**Soundness:** 3
**Presentation:** 4
**Contribution:** 3
**Rating:** 6
**Confidence:** 4

**Summary:**

This paper proposes an idea for obtaining LoRA adapters for few-shot style-conditioned generation. The basic idea is, instead of learning LoRA adapters from only a few examples, using a variational auto-encoder to generate the LoRA adapter. Empirical results demonstrate the capacity of the proposed algorithm.

**Strengths:**

- This paper is well written, and the technical details are well explained, in terms of how the proposed algorithm was implemented.
- The general idea makes sense, based on the reviewer's understanding of VAE and its capability.
- The evaluation was conducted on three different datasets, which demonstrate the applicability of the proposed algorithm.

**Weaknesses:**

- Although the proposed idea is sounding, the experiment results should be stronger to demonstrate the utility of the proposed algorithm. After all, with the availability of other large-scale pre-trained model ready to use, the benefit of using VAE most be significant to over shadow it computational cost.
- The empirical results will be more convincing if the proposed algorithm was demonstrated with multiple pre-trained models
- It is likely that I missed something from the paper, but I could not find the implementation details of the VAE in the main paper.
- Part of the paper needs to be further proofread, e.g., equation 6.

**Questions:**

Aligned with my comments in the **weaknesses** section, here are some questions

- Can you demonstrate more experiment results with other pre-trained models?
- What is the exact structure used as variational auto-encoder and decoder?

---

> ### Author Response · Authors · 2024-11-21
>
> Thank you for your insightful comments and feedback.
>
> Regarding the VAE structure, the encoder consisted of a fully connected layer of dimension (input_size, 512) followed by 2 more fully connected layers each of dimension (512, latent_dim) to output the mean and covariance vectors. The decoder consisted of 3 fully connected layers of dimensions (latent_dim, 512), (512, 512) and (512, input_dim). We used a latent dimension of 8 in this work. We'll add a figure with the VAE architectural details to the paper.
>
> We have more results related to the other questions that we'll add closer to the end of the discussion period.

---

### Author Response · Authors · 2024-11-28

We thank all the reviewers for their useful comments and feedback. In addition to the individual comments to the reviewers, we summarize the changes in the updated submission:
 - We've removed the original Figure 2 showing the LoRA extraction process, as per the request of reviewer qmsk and replaced it with a figure depicting the VAE architecture, as per the request of reviewer nsiM.
 - As per the request of all reviewers who asked for results on other pretrained models, we've added results from an experiment using GPT-2 as the pretrained model to the appendix, in section A.8.
 - As per the request of reviewer qmsk, we've added more details about the corpus size to section 3.1.1, some text related to performing one gradient update step in section 3.2 and more details about LoRA finetuning hyperparameters in section 4.2. Also per their request, we've performed an additional evaluation using the style embeddings from Wegmann et al., results of which we've added to the appendix in section A.9. Additionally, we've fixed the issue with equations 3-5 and bolded highest values in Table 1.
 - Per reviewer CGYD's comments, we've added a reference to "Extracting Latent Steering Vectors from Pretrained Language Models" to the Related Work. We've also fixed some of the notation and edited some of the writing, as per their suggestions.
 - We've added an explicit reference to the section of the appendix (A.4.3) that addresses the concerns regarding choice of finetuned models raised by reviewer mBaA. Also per their request, we've added a number of sample generations to the appendix, in section A.10.

---

### Meta-Review · Area_Chair_xjTR · 2024-12-05

**Metareview:**

### **Summary**
This paper presents a novel approach to adapting large language models (LLMs) for few-shot style-conditioned text generation. The method introduces a latent representation of LoRA weight deltas using a variational autoencoder (VAE). By interpolating within this learned latent space, the method can generate style-specific fine-tuned models with minimal data. Empirical evaluations demonstrate competitive performance against baselines across three datasets, particularly in low-resource settings.

### **Strengths**
1. **Innovative Approach**: The use of a VAE to encode LoRA weight deltas as a style representation is novel and has the potential to generalize to other style-conditioning tasks.
2. **Focus on Few-Shot Learning**: The proposed method addresses a key challenge in low-resource scenarios, offering a computationally efficient alternative to traditional fine-tuning or prompting.
3. **Empirical Validation**: Results across three datasets show clear benefits of the approach in very few-shot scenarios (<16 samples), with robust comparisons to baselines like prompting and direct fine-tuning.

### **Weaknesses**
1. **Limited Generalizability**:
   - The method was primarily evaluated on LLaMA-2 (7B) and GPT-2, with no evidence of scalability to larger or more diverse models.
   - Cross-domain applicability, such as style transfer tasks, was not explored, leaving the generalizability of the approach unclear.
2. **Theoretical Foundations**:
   - The choice of VAE filtering criteria and thresholds appears ad-hoc, with limited theoretical justification or ablation studies to validate these choices.
3. **Evaluation Gaps**:
   - The absence of human evaluations weakens the empirical claims, as style-conditioned text generation is inherently subjective.
   - Qualitative analyses of generated samples are minimal, and reviewers requested more detailed insights into where the method succeeds or fails.
   - Baselines like prompting outperform the proposed method in higher resource scenarios (e.g., 16+ samples), raising questions about its broader utility.
4. **Presentation Issues**:
   - While improved during the rebuttal, parts of the writing and figures remain unclear or inconsistent.
   - Equations and variable definitions (e.g., EQ 3-5) could be clearer to aid comprehension.


### **Recommendation**
While the paper introduces a novel and promising approach, its limitations in evaluation, theoretical grounding, and generalizability prevent it from meeting the high bar of ICLR. The lack of human evaluations and minimal qualitative analysis weaken the empirical contributions, and the scope of the experiments is too narrow to establish broad applicability. Future iterations could address these gaps to make a stronger case for acceptance.

**Additional Comments On Reviewer Discussion:**

The authors made substantial efforts to address reviewer concerns:
- Added evaluations on GPT-2 to demonstrate generalizability beyond LLaMA-2.
- Provided detailed VAE architecture and clarified the training process.
- Enhanced the presentation by fixing equations, improving tables, and replacing Figure 2 with a more relevant diagram.
- Included new analyses, such as comparisons with alternative style representations and ablations on fine-tuned model selection.

While these revisions improved the paper, some critical gaps remain unaddressed:
- No human evaluations or extensive qualitative analyses were added.
- Theoretical justifications for certain design choices (e.g., latent space construction) remain weak.
- Broader applicability (e.g., cross-domain evaluations or larger LLMs) was not sufficiently explored.

---

### Decision · Program_Chairs · 2025-01-22

Reject